# Revisiting Sparse Learning for Classification: Comprehensive Comparisons of $L_0$ and $L_1$ Approximations with Early Stopping

## Abstract

Understanding the comparative performance of $L_0$ and $L_1$ models is crucial for developing accurate and efficient machine learning systems, particularly in noisy, real-world settings. The current understanding in the literature is that $L_1$-penalized linear models perform better than $L_0$ models as noise increases. However, prior studies have largely relied on small and synthetic datasets and limited comparisons between differing optimizers, while leaving experimentation reflective of practitioner concerns underexplored. We fill these gaps in analysis by testing multiple different $L_0$ and $L_1$ approximate optimizers on a larger variety of real datasets and using a realistic workflow for a practitioner who at the end of the day values empirical out-of-sample performance. We demonstrate that empirical performance differences between $L_0$ and $L_1$ models depend significantly on the choice of optimizer and dataset characteristics. In many cases, the difference in performance by changing the optimization algorithm, while leaving the regularization penalty constant, is larger than the differences in changing the penalty. Together, our results show that $L_0$-penalized approximate optimizers with early stopping can remain competitive with $L_1$ models even for noisier datasets and are more viable than previously recognized.

## 1 Introduction

Methods for sparse regression and classification are useful for a multitude of reasons, especially when confronting problems with a large number of features. Induced sparsity can be important for reducing overfitting and improving model generalization on unseen data. The regularization reduces the variance of the model predictions, and this has been demonstrated to improve model generalization on real-world datasets. Moreover, sparsity can reduce required resources, and improve model interpretability. These are some of the reasons why methods for sparse linear and logistic regression are among the most commonly used tools in the toolbox for machine learning (Hastie et al., 2015).

The LASSO (Tibshirani, 1996) is a widely used and highly successful regularization method for regression and classification problems and induces both coefficient sparsity as well as coefficient shrinkage. Its wide and continued use has led to ongoing studies to implement efficient solvers (Massias et al., 2018).

If we denote by $f(\boldsymbol{\theta})$ the loss function of a regression or classification problem, the LASSO is given by

$$\min_{\boldsymbol{\theta}} \quad f(\boldsymbol{\theta}) + \lambda_1 \|\boldsymbol{\theta}\|_1. \tag{1}$$

Solutions to the optimization problem naturally induce sparsity – some subset of the coefficients $\boldsymbol{\theta}$ will be zero. In addition, the $L_1$ constraint (or penalty) on

$$\|\boldsymbol{\theta}\|_1 = \sum_{i=1}^{p} \|\theta_i\|_1 \tag{2}$$

induces shrinkage of the coefficient magnitudes.

The Best Subset Selection regularization scheme instead uses the $L_0$ pseudo-norm. Therefore, it induces sparsity but without any shrinkage of the coefficient magnitudes.

It is stated,

$$\min_{\boldsymbol{\theta}} \quad f(\boldsymbol{\theta}) + \lambda_0 \|\boldsymbol{\theta}\|_0 \tag{3}$$

where

$$\|\boldsymbol{\theta}\|_0 = \sum_{i=1}^{p} 1(\theta_i \neq 0). \tag{4}$$

Solving this exactly poses solving a mixed-integer optimization problem. Exact solutions for regression problems via the leaps and bounds algorithm (Furnival & Wilson, 1974) were available in the `leaps` and `bestglm` packages, but could only solve problems with $p \sim 30$ features. Recent advances enabled mixed-integer optimizers for regression problems such as (Bertsimas et al., 2016) to tackle problem sizes roughly having a number of samples $n \sim 10^2$ and a number of features $p \sim 10^4$.

There also exist first order methods (Blumensath & Davies, 2009; Bahmani et al., 2013) and second-order methods (Yuan & Liu, 2017; Zhou et al., 2021; Wang et al., 2021) for approximately solving $L_0$-regularized problems for regression and classification.

In this manuscript, we revisit and challenge earlier findings applicability to realistic workflows by comparing several $L_0$-regularized and $L_1$-regularized approximate solvers with early stopping on an extensive selection of datasets for binary classification, with varying amounts of feature and label noise. We demonstrate that the choice of optimizer can be equally as important as the choice of regularization class under large levels of noise, and certain $L_0$-regularized optimizers retain stable performance at moderate levels of noise.

## 2 Related Studies

Having coefficient shrinkage, the LASSO was believed to be superior to $L_0$-regularization for datasets for data with a lower signal-to-noise ratio (SNR) (Hastie, 2001). Some evidence was presented to this effect in previous studies for regression (Hastie et al., 2020) and classification (Dedieu et al., 2021). However, these studies primarily demonstrated results on simulated data and had very limited results on real datasets. Hastie et al. (2020) compared $L_0$-regularization and LASSO for regression problems, and only used the mixed-integer optimization method provided by (Bertsimas et al., 2016) for the $L_0$-regularized model. They concluded that LASSO gave better test accuracy in the low SNR regime, and worse accuracy in the high SNR regime, and the transition point in SNR depended on the problem dimensions: the number of training samples $n$ and number of features $p$. This work only studied regression problems, rather than classification problems which are the focus of this manuscript. Moreover, this work performed comparisons exclusively on simulated/synthetic data, where the underlying data-generating process is known.

On the other hand, Dedieu et al. (2021) studied binary classification problems, which are also the focus of this manuscript, and compared their optimizer designed to solve the combined $L_0 + \alpha L_q$ penalty with LASSO. They found that combined $L_0 + \alpha L_2$ penalty could outperform LASSO, with the $L_2$ penalty inducing coefficient shrinkage and reducing variance. Unfortunately, their study did not share any results for the pure ($L_0$-only) selector, which would have been highly relevant and informative. At a high level, their conclusions were largely similar to (Hastie et al., 2020), however, these conclusions were based mostly on simulated datasets with very limited real datasets. The simulated data were generated as multivariate Gaussian features with various correlation strengths. For comparison on real datasets, they showed only three (Arcene, Dexter, and Dorothea) taken from the NIPS 2003 Feature Selection Challenge (Guyon et al., 2004).

In contrast, in this study, we compare the performance of approximate $L_0$-regularized methods and $L_1$ methods for a wide set of binary classification problems. In addition, we compare a variety of optimizers within each regularization class. Additionally, we compare the empirical and practical performance of the methods on a wide variety of binary classification datasets from `https://www.csie.ntu.edu.tw/~cjlin/libsvmtools/datasets/binary.html` at variable SNR. We perform this extensive experimentation on real

datasets to give guidance that applies to real-world classification problems in which the ground truth data-generating process is not known. Additionally, these experiments are performed to mimic a typical machine learning workflow for real-world data: models are optimized on training data, hyper-parameters – including the optimizer termination criteria – are optimized on held-out validation data, and expected performance is estimated by performance on held-out testing data. Our experiments challenge the applicability to data science practice the prior consensus that the relative strengths and weaknesses of $L_0$-regularization and LASSO are mostly a function of the SNR and coefficient shrinkage. Moreover, by comparing multiple optimizers for $L_0$-regularization and LASSO, we also challenge the idea that the cost functions alone are predictive of performance as SNR is varied. Instead, we will show that differences between different approximate optimizers can be as significant and relevant in determining performance.

## 3 Experimental Method and Results

In this section, we present comparisons between variants of the Iterative Hard Thresholding (IHT) (Blumensath & Davies, 2009), with momentum (IHTM), and L0Learn (Dedieu et al., 2021; Hazimeh et al., 2023) $L_0$-regularized optimizers, the optimizer with mixed $L_0$ and $L_2$ penalty from L0Learn, and two LASSO optimizers: LIBLINEAR (Fan et al., 2008) and SAGA (Defazio et al., 2014), which was based on Stochastic Average Gradient (SAG) method (Schmidt et al., 2017).

In Iterative Hard Thresholding (IHT) (Blumensath & Davies, 2009): the weights are updated at each iteration by a projected gradient descent method,

$$\boldsymbol{\theta}_{t+1} = \Pi_k(\boldsymbol{\theta}_t - \eta \nabla_{\boldsymbol{\theta}} f(\boldsymbol{\theta}_t)), \tag{5}$$

where $\eta$ is a learning rate, and the operator $\Pi_k$ projects the weights onto the nearest point of the $L_0$ ball $|\boldsymbol{\theta}|_0 < k$. This projection is accomplished by sorting the weights $\boldsymbol{\theta}$ by their magnitude and keeping the $k$-largest while zeroing the rest:

$$\Pi_k(\boldsymbol{\theta}) = \boldsymbol{\theta}' \quad \text{where} \quad \theta_i' = \begin{cases} \theta_i & \text{if } |\theta_i| \geq |\theta_{[k]}| \\ 0 & \text{if } |\theta_i| < |\theta_{[k]}|, \end{cases} \tag{6}$$

where $\theta_{[k]}$ denotes the $k$-th largest element in the sorted list of $|\theta_i|$ values.

We also extend the Iterative Hard Thresholding method to include a proposal vector given by gradient descent with momentum (IHTM),

$$\boldsymbol{\theta}_{t+1} = \Pi_k(\boldsymbol{\theta}_t - \eta \boldsymbol{v}_t), \tag{7}$$

where

$$\boldsymbol{v}_t \equiv \beta \boldsymbol{v}_{t-1} + \nabla_{\boldsymbol{\theta}} f(\boldsymbol{\theta}_t)), \tag{8}$$

with momentum decay parameter $\beta = 0.9$.

We also compare the $L_0$-regularized optimizers described in (Dedieu et al., 2021) and implemented in (Hazimeh et al., 2023), which we refer to as L0Learn. The L0Learn optimizers use coordinate descent and local combinatorial search to approximately minimize its objective,

$$\min_{\boldsymbol{\theta}} \quad f(\boldsymbol{\theta}) + \lambda_0 \|\boldsymbol{\theta}\|_0 + \lambda_2 \|\boldsymbol{\theta}\|_2, \tag{9}$$

together with heuristics including correlation screening and greedy cycling order for computational efficiency.

Our focus is to compare $L_1$ to a method in $L_0$ selection as both are oriented towards some form of "minimum subset", and represent two widely studied and used approaches. While many interpolations of these exist, extensive numerical simulations require we limit ourselves from excessive additional penalties like the Elastic-Net (which intentionally will select correlated predictors) and $L_q$ penalties for $q \in (0, 1)$. We compare the L0Learn optimizer with both the pure best subset $L_0$ selector as well as a mixed selector which has both $L_0$ and $L_2$ penalties. We include the mixed $L_0$ and $L_2$ selector to make contact with the authors' existing study and recommendations.

For LASSO, we compare two optimizers: LIBLINEAR (Fan et al., 2008) and SAGA (Defazio et al., 2014). LIBLINEAR is a widely used library for large-scale linear classification. LIBLINEAR (Fan et al., 2008) solves $L_1$-regularized logistic regression problems using coordinate descent, where the objective is

$$\min_{\boldsymbol{\theta}} \ \frac{1}{n} \sum_{i=1}^{n} \log \left(1 + \exp(-y_i \cdot x_i^\top \boldsymbol{\theta})\right) + \mathcal{R}(\boldsymbol{\theta}),$$

with $\mathcal{R}(\boldsymbol{\theta}) = \lambda_1 \|\boldsymbol{\theta}\|_1$. The algorithm updates one coordinate $\theta_j$ at a time, holding the others fixed. The update takes the form:

$$\theta_j \leftarrow \text{sign}(z_j) \cdot \max\left(0, |z_j| - \lambda_1\right),$$

where $z_j$ is the coordinate-wise update derived from the gradient of the smooth loss. This soft-thresholding operation promotes sparsity and allows LIBLINEAR to scale efficiently to high-dimensional problems.

SAGA (Defazio et al., 2014) is a variance-reduced stochastic gradient method designed for finite-sum problems. At each iteration, SAGA samples a data point $i$ uniformly at random and computes the gradient of the loss $f_i(\boldsymbol{\theta}_t)$ on that example. It corrects this gradient using stored gradients from previous iterations to reduce variance, resulting in the update:

$$\boldsymbol{\theta}_{t+1} = \text{prox}_\eta^{\mathcal{R}} \left(\boldsymbol{\theta}_t - \eta \left(\nabla f_i(\boldsymbol{\theta}_t) - \nabla f_i(\boldsymbol{\phi}_i) + \bar{\boldsymbol{\alpha}}\right)\right),$$

where $\boldsymbol{\phi}_i$ is the previous iterate used to compute $\nabla f_i$, and $\bar{\boldsymbol{\alpha}}$ is the average of all stored gradients. The proximal operator is defined as

$$\text{prox}_\eta^{\mathcal{R}}(\boldsymbol{v}) = \arg\min_{\boldsymbol{\theta}} \left\{\frac{1}{2\eta} \|\boldsymbol{\theta} - \boldsymbol{v}\|_2^2 + \mathcal{R}(\boldsymbol{\theta})\right\},$$

and handles non-smooth regularization, in our case the $L_1$-penalty.

Table 1: All the datasets included in comparisons between $L_0$ and $L_1$ optimizers studied.

| Dataset | # Train | # Valid | # Test | # Features |
|---|---|---|---|---|
| a1a | 624 | 83 | 83 | 123 |
| arcene | 88 | 44 | 44 | 10000 |
| australian | 512 | 51 | 51 | 14 |
| breast-cancer | 362 | 58 | 58 | 10 |
| cod-rna | 31732 | 3979 | 3979 | 8 |
| colon-cancer | 34 | 5 | 5 | 2000 |
| dexter | 300 | 150 | 150 | 20000 |
| diabetes | 426 | 55 | 55 | 8 |
| dorothea | 156 | 34 | 34 | 100000 |
| german.numer | 482 | 59 | 59 | 24 |
| gisette | 4786 | 593 | 593 | 5000 |
| heart | 198 | 21 | 21 | 13 |
| ijcnn1 | 9706 | 8712 | 8712 | 22 |
| ionosphere | 196 | 28 | 28 | 34 |
| leukemia | 22 | 14 | 14 | 7129 |
| liver-disorders | 110 | 100 | 100 | 5 |
| madelon | 2000 | 300 | 300 | 500 |
| phishing | 7884 | 956 | 956 | 68 |
| sonar | 162 | 16 | 16 | 60 |
| splice | 966 | 1044 | 1044 | 60 |
| svmguide1 | 2178 | 2000 | 2000 | 4 |
| w1a | 144 | 1407 | 1407 | 300 |

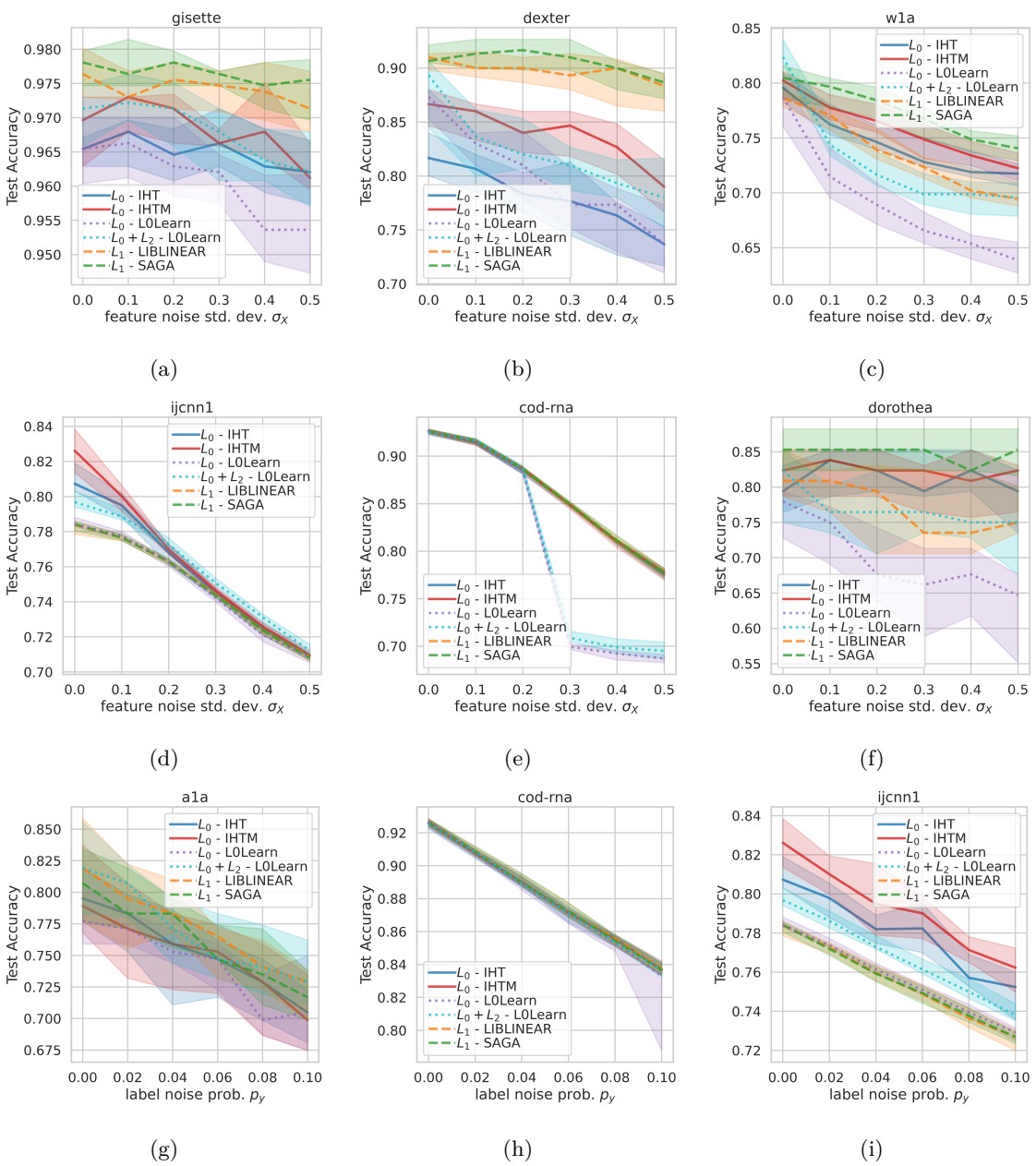

Figure 1: Select comparisons of $L_0$ Best Subset Selection and $L_1$ LASSO optimizers on a subset of the datasets studied.

The performance of the optimizers is compared on a wide variety of binary classification datasets from `https://www.csie.ntu.edu.tw/~cjlin/libsvmtools/datasets/binary.html` as well as three datasets from the NIPS 2003 feature selection challenge, that are all listed in Table 1. The datasets are first balanced by label via undersampling, and the number of training, validation and test samples in Table 1 are after balancing. We add noise to the data by two different methods which are plotted separately:

1. we add Gaussian noise with standard deviation $\sigma_X$ to the normalized features of each dataset, allowing us to explicitly vary the amount of noise present in the features not associated with the true label, or

2. we flip the binary labels in the dataset with probability $p_y$.

These random sources of noise are sampled randomly twenty times. The performance of the optimizers on the original datasets without added noise is displayed in each figure at the points for which $\sigma_X = 0$ and $p_y = 0$, which is when no feature noise or label noise is added.

We hold out separate validation and test data from the training data. Following a practical/real-world scenario, in which a practitioner's primary concern is out-of-sample test performance, each model's hyper-parameters are optimized on the validation data with fifty trials of optuna (Akiba et al., 2019).

The hyperparameters include the following: For all methods, the termination conditions are hyperparam-eterized and optimized. These are the maximum number of iterations, with a ceiling of $1,000$ iterations, and a convergence tolerance, with a floor of $10^{-8}$ and a ceiling of $10^{-2}$. For the IHT, IHTM, and L0Learn methods the integer $k$, the allowed number of nonzero weights, with a floor of 1 and ceiling of the number of dataset features, for the LIBLINEAR and SAGA methods the real numbers $\lambda_1$ for the LASSO penalty. For the IHT and IHTM methods, the learning rate has a floor of $10^{-4}$ and a ceiling of $10^{-1}$.

We note importantly that the convergence tolerance parameter is calculated differently by different opti-mizers. For example, LIBLINEAR checks if the violation of the Karush-Kuhn-Tucker conditions is below a tolerance (Fan et al., 2008). The original SAGA manuscript (Defazio et al., 2014) does not specify an early stopping condition, however, the implementation in scikit-learn checks if the maximum relative change in any of the model weights falls below a tolerance parameter. On the other hand, L0Learn checks if the relative change in the objective function falls below a tolerance parameter. Finally, the IHT and IHTM methods check if the relative change in the objective function falls below a tolerance parameter. Given that all of these optimizers implement different stopping conditions, the only way to compare them in a manner that is relevant to the practitioner is to let these criteria be independently optimized by the hyperparameter optimization process on validation data. Ultimately, a practitioner does not care to what degree the opti-mization has converged to a solution for the *training data* objective, but instead, values the performance on unseen test data.

For these figures, the medians are plotted as lines with the interquartile ranges shaded. The `gisette` (Fig. 1a) and `dexter` (Fig. 1b) datasets exhibit the behavior of $L_0$-regularization compared to LASSO that we would expect based on prior studies as feature noise increases. Namely, for these datasets, there is clear degradation in the performance of *all* $L_0$-regularized methods as noise is increased, significantly slower degradation in the performance of the LASSO methods, and both LASSO methods outperform all $L_0$-regularized methods at large values of noise.

However, for the `w1a` (Fig. 1c) dataset, we see behavior that was not anticipated in previous studies; as feature noise increases, the performance of the L0Learn optimizer on test data degrades more rapidly than the other optimizers. This is a case where the optimizer makes a larger difference in test performance than the nominal regularization penalty. The `ijcnn1` (Fig. 1d) dataset also exhibits behavior not explained in previous studies; for even the largest values of added feature noise, the LASSO methods do not overtake the $L_0$-regularized methods in performance.

For the `cod-rna` (Fig. 1e) the performance of all methods degrades systematically at large levels of added noise. There is no performance gap between the LIBLINEAR and SAGA $L_1$ methods and the IHT and IHTM $L_0$ methods as feature noise increases. However, both of the L0Learn optimizers perform much worse - again, a case where the optimizer yields a much larger difference in test performance than the regularization penalty. Finally, for the `dorothea` (Fig. 1f) the performance of both the IHT and IHTM methods remains stable as feature noise is increased.

Now, we discuss the behaviors on individual datasets as the label noise is increased. For certain datasets, such as `a1a` (Fig. 1g) and `cod-rna` (Fig. 1h) datasets among others, as label noise increases, all models degrade in test performance. However, for these datasets, no hierarchy develops of $L_1$ over $L_0$ models. For the `ijcnn1` (Fig. 1i) dataset, the performance gap between the L0Learn $L_0$ optimizer performs very similarly to the LIBLINEAR and SAGA $L_1$ optimizers, with a systematic gap in performance to the IHT and IHTM $L_0$ optimizers. This is again a case where the differences between optimizers of the same regularization class are the more significant effect.

From these comparisons, we find the absence of a simple pattern or 'story' between $L_0$-regularization and LASSO as it pertains to the effect of coefficient shrinkage and data noise; on the contrary, the performance differences among the different $L_0$-regularized optimization methods IHT, IHTM, and L0Learn, or between LIBLINEAR and SAGA LASSO optimizers, are often as large or larger than the differences between $L_0$ and $L_1$ methods.

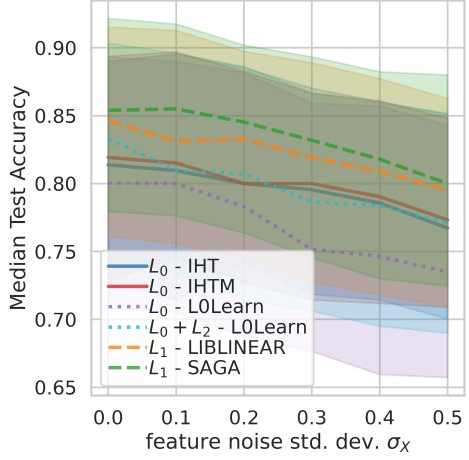

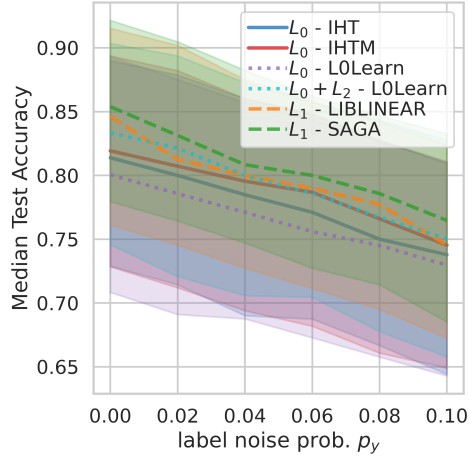

(a) Test accuracy as a function of added feature noise.

(b) Test accuracy as a function of added label noise.

Figure 2: The test accuracy performance (the median taken over datasets) for each optimizer as a function of added feature noise (a) and label noise (b).

In Fig. 2a we show plots demonstrating the performance of each optimizer over all datasets studied as feature noise is varied. At the very largest levels of added feature noise ($\sigma_X = 0.5$), there is weak evidence to the previous understanding. That is, both LIBLINEAR and SAGA $L_1$ optimizers have test performance which is systematically slightly higher than all of the $L_0$ optimizers, including L0Learn's mixed $L_0 + L_2$ penalty. However, these are not statistically significant given the large interquartile spread across datasets.

However, the fact that the $L_0 + L_2$ penalty (with coefficient shrinkage) does not systematically improve performance at the largest levels of noise with respect to all pure $L_0$ methods is surprising. This is contrary to the L0Learn GitHub page, which 'strongly recommends' using the mixed penalty, justified by the same concerns regarding SNR and overfitting without shrinkage (Hazimeh et al., 2023). In Fig. 2b we show plots demonstrating the performance of each optimizer over all datasets studied as label noise is increased. The performance of all methods systematically degrades with increased label noise, but again with no statistically significant performance gap between $L_0$ and $L_1$ methods.

In Fig. 3a we show plots demonstrating the sparsity, measured by the number of nonzero weights, of each optimizer over all datasets studied as feature noise is varied. Overall, we see that $L_0$ methods tend to produce sparser (fewer nonzero weights) models than both $L_1$ methods. The sparsest models are produced by the L0Learn $L_0$ optimizer. But even the Iterative Hard Thresholding method produces sparser models than either $L_1$ solver.

In Fig. 3b we show plots demonstrating the sparsity, measured by the number of nonzero weights, of each optimizer over all datasets studied as label noise is varied. Overall, the situation is similar to the previous one. $L_0$ methods are sparser on average, with L0Learn's optimizer producing the sparsest models.

In Fig. 4a we plot the relative (dis)advantages of LASSO methods and $L_0$-regularized methods from a typical model selection perspective. This is calculated as follows. For each dataset, we take the top-performing $L_0$ method by validation accuracy and the top-performing $L_1$ method by validation accuracy. The test accuracy and number of nonzero weights (NNZ) for each are saved. Then we compute ratios: the $L_1$ sparsity

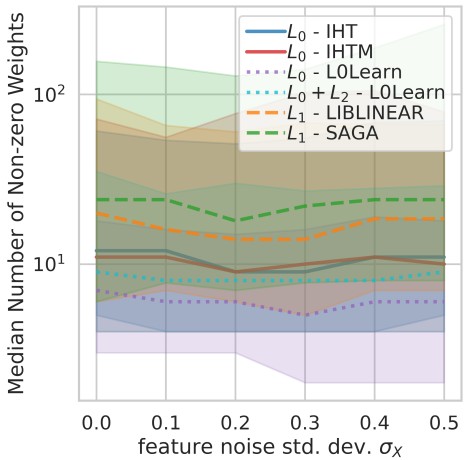
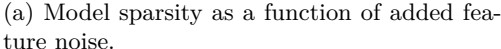
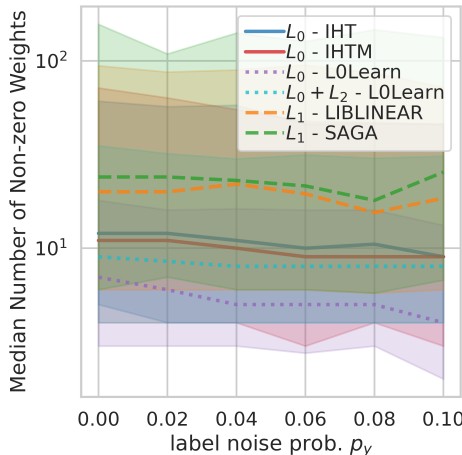

(a) Model sparsity as a function of added feature noise.

(b) Model sparsity as a function of added label noise.

Figure 3: Model sparsity (measured by the number of nonzero weights, median over datasets) and optimizer performance as a function of added feature noise (a) and label noise (b).

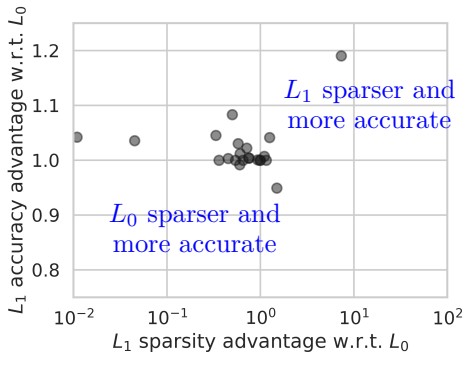
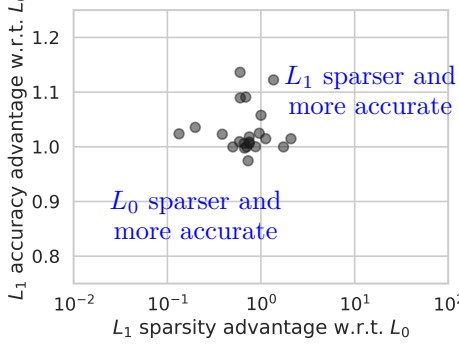

(a) Datasets without added noise.

(b) Datasets with the largest added feature noise.

Figure 4: Comparison of the best-performing $L_0$ and $L_1$ optimizers in terms of sparsity and test performance. (a) Results for datasets without added noise. (b) Results for datasets with the largest amount of added feature noise.

advantage w.r.t. $L_0$ is given $\text{NNZ}_{L_0}/\text{NNZ}_{L_1}$ (less nonzero weights is more sparse and more desirable), and the $L_1$ accuracy advantage w.r.t. $L_0$ is given Test Acc.$_{L_1}/$Test Acc.$_{L_0}$. We can read this plot as follows: In the left half, the best $L_1$ method produces a *less sparse* model than the best $L_0$ method, while in the right half, it produces a *more sparse* model than $L_0$. In the top half, the best $L_1$ method is *more accurate* than the best $L_0$ method, while in the bottom half, it is *less accurate.*

We see that, for some datasets, the best $L_1$ method is slightly more accurate and slightly less sparse than the best $L_0$ method, and for many datasets, the accuracy performance is quite similar. Similarly, in Fig. 4b we plot the relative (dis)advantages of LASSO methods and $L_0$-regularized methods from a typical model selection perspective, but now on the datasets with the largest amount of added feature noise $\sigma_X = 0.5$. Interestingly, in this case, where the data have a large amount of feature noise, the LASSO tends to be more accurate but usually still at the expense of being less sparse. So although both sparsity and coefficient

shrinkage are sources of regularization, the best-performing LASSO method usually favors more nonzero coefficients with smaller magnitudes rather than fewer nonzero coefficients with larger magnitudes.

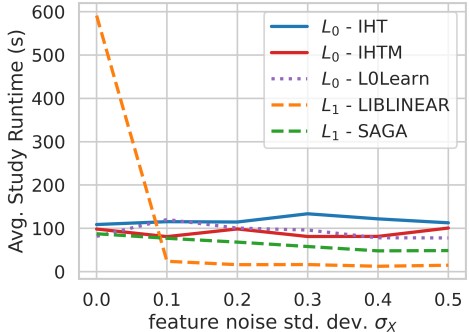
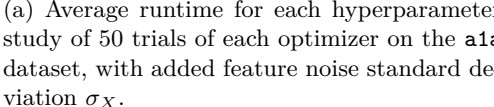

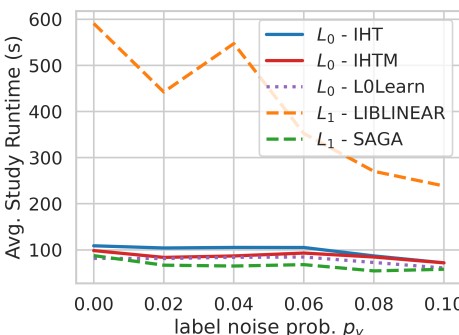

(a) Average runtime for each hyperparameter study of 50 trials of each optimizer on the `a1a` dataset, with added feature noise standard deviation $\sigma_X$.

(b) Average runtime for each hyperparameter study of 50 trials of each optimizer on the `a1a` dataset, with added label noise with probability $p_y$.

Figure 5: Average runtime for each hyperparameter study of 50 trials of each optimizer on the `a1a` dataset as a function of added feature noise (a) and label noise (b).

Finally in Fig. 5 we compare the average runtime for a single hyperparameter optimization experiment of 50 trials on the `a1a` dataset across the optimizers. We choose to measure the optimizer runtime performance in this manner to simulate the compute cost a practitioner (who must perform such a study because a priori they do not know the optimal hyperparameters) would require. Each hyperparameter experiment is performed on a single Intel Xeon CPU E7-8890 v4 @ 2.20GHz processor. The average runtime for each experiment is similar for all optimizers except LIBLINEAR, which is roughly six times more expensive for the original dataset. As noise is added to the features, the LIBLINEAR termination conditions trigger quickly, leading to a runtime that is faster than all others. As the amount of label noise is added, however, the LIBLINEAR optimizer remains between two and six times more expensive than the rest.

## 4 Conclusion

This manuscript provides a thorough evaluation of sparse learning techniques, challenging the applicability of common assumptions about LASSO and Best Subset Selection approximations across real-world datasets. Our experiments show that the strengths and weaknesses of these methods vary significantly, especially under different noise levels.

A key insight is that optimizer-specific behaviors can heavily influence performance, both in terms of test accuracy and model sparsity, sometimes more than the choice between $L_1$ and $L_0$ regularization. This underscores the importance of considering the interactions of both the regularization and the optimization strategy with noise in the data in practice.

Our results indicate that the traditional view linking sparse learning performance primarily to SNR and coefficient shrinkage may be too simplistic to provide guidance applicable to practitioners operating on real noisy data. The choice of optimizer plays a critical role, with varied performance under the same noise conditions. Typically, when the best $L_1$ optimizer is more accurate, it is also less or equivalently sparse. This study offers actionable insights through a thorough comparison across multiple datasets and optimizers.

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
