# OpenReview forum: "Revisiting Sparse Learning for Classification: Comprehensive Comparisons of $L_0$ and $L_1$ Approximations with Early Stopping"
_TMLR — Rejected by TMLR_

### Review · Reviewer_Vq3q · 2025-03-22

**Summary Of Contributions:**

This paper compares the performance of the lasso against non-convex $L_0$
penalty methods for variable selection in logistic regression problems.
The authors focus their attention on real datasets from LIBSVM, use a
realistic train-validate-test pipeline, and consider the effects of different
optimizers on the final test accuracy.
Unlike previous work, their experiments show that the lasso does not always
outperform $L_0$ methods when the signal-to-noise ratio (SNR) is high.
Instead, they obtain mixed results where the gap between different optimizers
is often larger than the gap between methods with the $L_1$ or $L_0$ penalty.
They also show that even when the lasso has better test performance, it
it obtains less sparse solutions than the $L_0$ penalty.

**Audience:**

Yes

**Broader Impact Concerns:**

None.

**Claims And Evidence:**

No

**Requested Changes:**

- Please report the hyper-parameter settings for the different optimizers, including their termination conditions. Do the final lasso solutions have the same training fits and $L_1$ norms? If not, they are not optimal solutions.

- Please include a plot of the final regularization strengths to accompany Figure 3. Otherwise we cannot draw any meaningful conclusions from this experiment.

- All figures reporting median accuracy should also include interquartile ranges so that the readers can understand the amount of variation in the results.

- I strongly feel that ground-truth solutions to the $L_0$-penalized problems obtained by solving the MIP should be included. Otherwise we cannot judge the quality of the approximate solution due to L0Learn and IHT.

**Strengths And Weaknesses:**

**Strengths**:

- Comparing $L_0$ and $L_1$ penalty methods on real datasets is much more
    interesting than on synthetic problems.

- Studying the variation due to different solvers captures an important part of
  real-world usage which is often ignored.

**Weaknesses**:

- Comparing different optimizers requires similar convergence criteria, but
  optimizer parameters, including termination conditions, are not discussed.

- Approximate solvers for the $L_0$ penalty aren't compared to exact
    mixed-integer program (MIP) solvers.

- Black-box hyper-parameter optimization makes it hard to draw conclusions
    about the relationship between SNR and sparsity.

- Mixed empirical results prevent drawing any concrete conclusions from
    the paper.

### Additional Details

I like the idea of a large scale comparison between $L_0$ and $L_1$ penalty
methods on real data. Advances in commercial MIP solvers means that exact $L_0$
penalty methods are more practical than ever before. Since the lasso can be
viewed as a convex relaxation of the $L_0$ penalty method, it makes sense
to revisit their pros and cons for binary classification. With that said,
I have several major concerns about this paper.

**Comparing Lasso Optimizers**:
In general, the solution to the lasso problem is not guaranteed to be unique
[2] and so it makes sense to compare the _test performance_ of different
solutions.
However, it's only interesting compare test performance, since different lasso
solutions must have the same fit on the training set and the same $L_1$ norm
[2].
Since different optimizers may select different solutions through
"implicit regularization" (see, for example, Gunasekar et al. [3]),
it is reasonable to compare the test performance of different solutions using
optimizers as a selection rule.
This is particularly true since this is typically how practitioners will
encounter different solutions.

This approach is complicated by the fact that numerical optimizers cannot
provide exact solutions to the lasso problem.
As a result, it is only sensible to compare optimizers if you ensure that they
terminate with (a) very accurate solutions that (b) have similar violation of
the KKT conditions or primal-dual gap.
If you use different termination conditions or the violations are different,
then you cannot guarantee test-time variation is due to the choice of optimal
solution rather than poor optimization.
As it is, the authors provide no information about optimization accuracy,
termination conditions, or optimizer hyper-parameter selection.
Thus, I cannot place any confidence in the conclusions about variation due
to lasso optimizers.

**Approximate Solvers for the $L_0$ Problem**

We encounter a similar problem for the solvers to the $L_0$ penalized problem.
Since the $L_0$ penalty is discontinuous and leads to a non-convex problem,
the authors consider several approximation methods.
It is reasonable to compare the performance of these approximations if you again
ensure that the optimizers are run to sufficient accuracy and have comparable
termination conditions.
However, the authors again provide no hyper-parameter details.

The authors also do not a discuss a critical aspect of using approximation methods:
you cannot draw conclusions about the true $L_0$-penalized solution, but only
about the solutions output by the approximate solver.
For example, we cannot conclude that solutions to $L_0$ penalized problems
perform worse on `luekemia` than the lasso; instead, we must say that solution
returned by IHT performs worse.
The only way to address this problem is to exactly solve the MIP using branch-
and-bound methods as done by Hastie et al. (2020).
This provides a ground-truth baseline that can be used to evaluate the quality
of approximations from IHT/L0Learn and to compare against the lasso.
I strongly suggest the authors add such baselines to the paper.

**Black-Box Hyper-Parameter Optimization**

In Figure 3, the authors try to elucidate a relationship between
model sparsity, noise level, and model/optimizer.
However, the authors do not report the final regularization strength selected
by their black-box hyper-parameter search procedure.
Since regularization strength controls the level of sparsity in both the
lasso and $L_0$-penalized problems, the data in Figure 3s are actually badly
confounded.
I personally think think that almost no conclusion can be drawn from this
figure without also reporting the regularization strength and giving a precision
description of how hyper-parameters were selected.

**Mixed Results**

Mixed experimental results are not necessarily a weakness.
However, I suspect that the authors obtain their mixed results
(compared to the synthetic regression setting) because of their experimental
procedure.
Failing to control (or at least report) the optimization accuracy, using
approximate solvers for the $L_0$ problem, and using black-box hyper-parameter
optimization introduce many confounding variables which could be causing the
inconsistency with previous work.
The authors need to be much more transparent about their experimental
procedure, report all of their hyper-parameters, and, ideally, release their
code.

### Minor Issues:

- Eq. (3): This isn't the Lagrange dual. It's the Lagrangian evaluated at the
dual-optimal parameter. If $p^*$ is the optimal value of Eq. (1), then this is,

$$p^* = \min_{\theta} \max_{\lambda \geq 0} f(\theta) + \lambda ( \|\theta\|_1 - \kappa).$$

where $\lambda_1$ is the maximizer in your notation. The Lagrange dual problem is,
$$
\max_{\lambda \geq 0} d(\lambda) := \min_{\theta} f(\theta) + \lambda (\|\theta\|_1 - \kappa),
$$
where it's clear that computing the dual function $d$ is as hard as solving the
Lasso problem.
Note also that convexity is not sufficient for Eq. (3) to be equivalent to Eq.
(1); you also need strict satisfiability (also called Slater's condition) for
strong duality to hold and for the dual problem to admit the solution
$\lambda_1$.
Slater's condition is trivially satisfied here as long as $f$ is nice.

- Eq. (6): Did you mean this to be $k$ instead of $\kappa$?

- Eq. (8): Similar to Eq. (3), this isn't the Lagrange dual problem for Eq. (5).

- Related work: It would be interesting to know if Dedieu et al. (2021) also
    compared to the elastic net ($L_1 + \alpha L_2$ penalty) and how this
    compared.

- Page 4: "that are diamonds" --- you can use the "unicode-math" package along with
    "\mdblkdiamond" to get a filled diamond symbol.

- Page 6: It's a little confusing to have a single line of text between
    Figures 2 and 3.

- You might also consider the Celer optimizer [1].
    However, comparing against different optimizers is only interesting
    if (a) you are running them under a fixed budget so that optimization
    speed is important or (b) they have different implicit regularization.
    In other words, running several optimizers which all converge to the
    max-margin classifier until convergence will yield the same train/test
    performance every time.

- Page 4: "the divergence between the performance of the two LASSO optimizers
  grows significantly" --- this is only notable if the two solvers satisfy the
  same sub-optimality/termination criteria when stopped.

- Figure 2: It's strange that nearly all methods improve in test performance
  when the feature or label noise is initially added. Indeed, the noise appears
  to acts as a type of regularization. Does this imply that most methods are
  under-regularized (and thus over-fitting) in the noiseless setting? I would
  expect regularized methods with the optimal regularization constant to only
  decrease in test accuracy as the level noise is increased.

- Have to checked to see if the Lasso solution is unique at the given
  regularization? It's unique when the data satisfy general position.
  See Tibshirani [2] for a detailed discussion.

- Figure 3: It's hard to draw any conclusions from this figure because the
    regularization constants are being chosen by a black-box hyper-parameter
    search for each noise level.

- If you report median results over many experiments, then you should also
    report the interquartile range. That way the reader has a sense for the
    amount of variation in the experiments.

### References

[1] Massias, Mathurin, Alexandre Gramfort, and Joseph Salmon. "Celer: a fast
solver for the lasso with dual extrapolation." International Conference on
Machine Learning. PMLR, 2018.

[2] Tibshirani, Ryan J. "The lasso problem and uniqueness." Electronic Journal
of Statistics 7 (2013): 1456-1490.

[3] Gunasekar, Suriya, et al. "Implicit regularization in matrix
factorization." Advances in neural information processing systems 30 (2017).

---

> ### Author Response · Authors · 2025-04-25
>
> 1. Thank you for pointing out this concern.
> We fully agree with your assessment; to address it, we changed and reran our experimental pipeline, producing the new plots in the revised manuscript. Previously we terminated each of the optimizers when their individual tolerance fell below $10^{-4}$. As you noted, LIBLINEAR checks the KKT conditions, and this defines the tolerance parameter. On the other hand, L0Learn checks if the relative change in the objective function falls below a tolerance, defining its parameter. As you stated, keeping these tolerances the same numerically does not make the termination of different optimizers `apples-to-apples' in our previous experiments.
> In the revised experiments and manuscript, the tolerance parameter for each optimizer is itself optimized on the validation dataset, following the practitioner's workflow. We have also added a paragraph describing this and all the hyperparameters in more detail in sec. 3 of the revision to address your concerns.
>
> 2. Yes, we agree, and thank you for pointing this out.
> Please see our previous reply regarding the revisions describing the experiments.
> We fully agree that we can not make claims about the true $L_0$ solutions since all of our solvers studied are approximate.
> Based on your feedback, we investigated running the L0Learn solver with local combinatorial search, but this solver was estimated to be more than 10x more expensive to perform on our experiments and was not able to finish under our current compute and time constraints. This was the only other solver for classification of which we were aware.
> The MIP solver discussed in Bertsimas et al 2016 and Hastie et al 2020 (https://github.com/ryantibs/best-subset/) is for regression problems, not the classification problems which we are studying in this manuscript.
>
> 3. Thank you for this point. In the revision, we have added content to section 3 to clarify the hyperparameters and their optimization.
>
> 4. Please see our previous replies regarding the revised description of the hyperparameter optimization. We will also release the code at camera-ready.
>
> 5. Thank you - we have revised the description.
>
> 6. Thank you - it was revised.
>
> 7. Thank you - it was revised.
>
> 8. Dedieu et al. (2021), whose optimizer we call L0Learn, does not include an elastic net option.
>
> 9. Thank you for this tip! We have revised the plot style to no longer use markers but linestyles and shaded interquartile ranges.
>
> 10. Thank you - it is no longer the case in the revision.
>
> 11. Thank you for bringing Celer to our attention. We have added a citation to Celer in the introduction of our revision.
>
> 12. This has been removed in the revision. Thank you.
>
> 13. After addressing the hyperparameter procedure w.r.t. stopping condition and running the experiments (please see reply to your first point) this is no longer the case.
>
> 14. We have not been able to perform this check.
>
> 15. Thank you - we do not intend Fig 3 to contain any information or theoretical insight w.r.t. the causes of sparsity. As you have said, there are many affects that can be contributing.  Fig 3 and all of our figures are intended to provide empirical information about what a practioner can expect regarding applying L0 and L1 methods to their problem.
>
> 16. Thank you for this suggestion - we have plotted interquartile ranges in the revision.

---

> ### Comment · Reviewer_Vq3q · 2025-04-28
>
> Many thanks for posting the revision and for replying to my concerns.
>
> 1. Treating the termination conditions as hyper-parameters and optimizing over them using validation loss is not at all the solution for which I was advocating in my review. My point was that different Lasso solvers will return different solutions to the Lasso problem and comparing the test performance of these solutions is interesting and missing from the literature. However, some care was required to ensure different solutions were indeed comparable. By optimizing over termination conditions, you are further complicating the comparison --- now "solutions" may not even be close to optimal for the Lasso problem. Indeed, the experimental results from this procedure have little to do with Lasso solutions, but now only apply to models obtained using this complex hyper-parameter tuning setup. You claim that this is justified because it "compare[s] them in a manner that is relevant to the practitioner" but I strongly disagree. Practitioners do not have the compute to run each optimizer at least fifty times in order to match your experimental setup and make your results relevant to their setting. We use models with simple mathematical descriptions like the Lasso exactly to have a reproducible reference point for theoretical and empirical results.
>
> 2. All of the my comments above apply to the L0 setting with the added confounder that L0Learn and IHT are only approximate solvers. I agree that the method from Bertsimas et al. (2016) doesn't apply to non-quadratic loss, but it is still possible to do best sub-set selection using brute-force enumerate. That would provide an interesting comparison point on the smallest datasets.
>
> While I appreciate the authors effort in providing a new revision, I think their changes actually reduce the applicability of the empirical results and worsen the paper. I am completely unsurprised that they "find the absence of a simple pattern or ‘story’ between L0-regularization and LASSO". Extensive hyper-parameter optimization leads to optimization bias and even if clear results were available, I strongly doubt if they would actually apply to either L0 or L1 regularized methods.

---

> ### Author Response · Authors · 2025-04-28
>
> Thank you for your reply.
>
> To clarify, a practitioner with a real dataset would not be adding 11 kinds of noise to their data, each sampled 20 times (this is where the 240,00 number in our earlier reply arises), etc.
> They would have one dataset. So the appropriate number to compare is how many optuna trials we run to optimize the hyperparameters jointly on *each dataset*, which is only 50 trials - certainly a reasonable number for a practitioner.
>
> By `practitioner' we mean one who is uninterested in the theoretical convergence of this or that optimizer/algorithm on a particular training dataset, and who is instead interested in empirical test set performance. Early stopping is a very commonly employed regularization.
> Our paper is focused on how the optimizers compare in this practitioner's setting.

---

> > ### Comment · Reviewer_Vq3q · 2025-04-28
> >
> > > So the appropriate number to compare is how many optuna trials we run to optimize the hyperparameters jointly on each dataset, which is only 50 trials - certainly a reasonable number for a practitioner.
> >
> > I agree that the 240,000 was taken out of context and is unfair. I've edited my comment to use the correct number.
> >
> > > Early stopping is a very commonly employed regularization. Our paper is focused on how the optimizers compare in this practitioner's setting.
> >
> > Early stopping prevents locks you in to a very poorly defined mathematical model and prevents any of your results from applying to standard L0 or L1 regularized problems. If you want to change your paper title to "Comprehensive Comparisons of Best Subset Selection and LASSO with _Early Stopping_", then that is valid. However, I don't agree that your current experimental setup has direct consequences for our understanding of either the Lasso or L0-regularization.

---

> ### Author Response · Authors · 2025-04-28
>
> Thank you for your clarification.
> Would you propose we change the the title to include 'with Early Stopping'
> and language in abstract, intro, conclusions, etc. from 'solvers' to 'approximate solvers'?
> Blumensath et al 2007 showed that iterative hard thresholding has fixed points which are local minima of the L_0 regularized loss. Given (as you point out) that we may have an early stopping prior to reaching the fixed point, we can also qualify mentions of the L_0 optimizers to say 'with early stopping', and similarly for L_1 optimizers.
> Please let us know if these are the changes you think are appropriate and we can provide an updated revision.

---

> > ### Comment · Reviewer_Vq3q · 2025-04-30
> >
> > I'd like to make it clear that I think treating the termination conditions as a hyper-parameter and optimizing over them has weakened the paper. Of course, I understand that you may disagree and prefer this approach. In that case, I think it is appropriate to qualify the claims in paper to say "with early stopping" as you suggest. Changing the title is also a good idea and would help better position this paper in the literature.

---

> > > ### Author Response · Authors · 2025-04-30
> > >
> > > Thank your for your reply.
> > > In that case, we will make the following changes in the revision:
> > >
> > > Title changes:
> > >
> > > Revisiting Sparse Learning for Classification: Comprehensive Comparisons of $L_0$ and $L_1$ Approximations with Early Stopping
> > >
> > >
> > > Abstract changes:
> > >
> > > Understanding the comparative performance of $L_0$ and $L_1$ models is crucial for developing accurate and efficient machine learning systems, particularly in noisy, real-world settings.
> > > The current understanding in the literature is that $L_1$-penalized linear models perform better than $L_0$ models as noise increases.
> > > However, prior studies have largely relied on small and synthetic datasets and limited comparisons between differing optimizers,
> > > while leaving experimentation reflective of practitioner concerns underexplored.
> > > We fill these gaps in analysis by testing multiple different $L_0$ and $L_1$ approximate optimizers on a larger variety of real datasets and using a realistic workflow for a practitioner who at the end of the day values empirical out-of-sample performance.
> > > We demonstrate that empirical performance differences between $L_0$ and $L_1$ models depend significantly on the choice of optimizer and dataset characteristics.
> > > In many cases, the difference in performance by changing the optimization algorithm, while leaving the regularization penalty constant,
> > > is larger than the differences in changing the penalty.
> > > Together, our results show that $L_0$-penalized approximate optimizers with early stopping can remain competitive with $L_1$ models even for noisier datasets and are more viable than previously recognized.
> > >
> > >
> > > Introduction changes:
> > >
> > > In this manuscript, we revisit and challenge earlier findings applicability to realistic workflows by comparing several $L_0$-regularized and $L_1$-regularized approximate solvers with early stopping on an extensive selection of datasets for binary classification, with varying amounts of feature and label noise.
> > > We demonstrate that the choice of optimizer can be equally as important as the choice of regularization class under large levels of noise, and certain $L_0$-regularized optimizers retain stable performance at moderate levels of noise.
> > >
> > > Related Studies changes:
> > >
> > > In contrast, in this study, we compare the performance of approximate $L_0$-regularized methods and $L_1$ methods for a wide set of binary classification problems.
> > > In addition, we compare a variety of optimizers within each regularization class.
> > > Additionally, we compare the empirical and practical performance of the methods on a wide variety of binary classification datasets from \url{https://www.csie.ntu.edu.tw/~cjlin/libsvmtools/datasets/binary.html}
> > > at variable SNR.
> > > We perform this extensive experimentation on real datasets to give guidance that applies to real-world classification problems in which the ground truth data-generating process is not known.
> > > Additionally, these experiments are performed to mimic a typical machine learning workflow for real-world data:
> > > models are optimized on training data, hyper-parameters -- including the optimizer termination criteria --
> > > are optimized on held-out validation data, and expected performance is estimated by performance on held-out testing data.
> > > Our experiments challenge the applicability to data science practice
> > > the prior consensus that the relative strengths and weaknesses of $L_0$-regularization and LASSO are mostly a function of the SNR and coefficient shrinkage.
> > > Moreover, by comparing multiple optimizers for $L_0$-regularization and LASSO, we also challenge the idea that the cost functions alone are predictive of performance as SNR is varied. Instead, we will show that differences between different approximate optimizers can be as significant and relevant in determining performance.
> > >
> > > Conclusion changes:
> > >
> > > This manuscript provides a thorough evaluation of sparse learning techniques, challenging the applicability of common assumptions about LASSO and Best Subset Selection approximations across real-world datasets. Our experiments show that the strengths and weaknesses of these methods vary significantly, especially under different noise levels.

---

### Review · Reviewer_FNTA · 2025-04-01

**Summary Of Contributions:**

This paper compares the $L_0$ and $L_1$ regularization in the classification
problems. It implements the $L_0$ regularization by four different optimizers
and the $L_1$ regularization by two different optimizers. Various real data are
considered in the comparisons.

**Audience:**

Yes

**Broader Impact Concerns:**

No concerns noticed.

**Claims And Evidence:**

Yes

**Requested Changes:**

1. Please fix the weaknesses pointed out earlier.
2. Since data with added noise are not real data anymore, please make it clear
   that when $\sigma_x=0$ and $p_y=0$, the results are really for real data
   performance.
3. I printed the paper and was not able to distinguish the lines in the
   figures. Please use different line types and/or markers so that people can
   still read the paper in grayscale.
4. Add a title for figure 1.
5. Figures 1 and 2 use different markers for the same optimizers. Please make
   the presentation consistent.
6. Define NNZ$_{L_0}$, NNZ$_{L_1}$, and possibly other notations.
7. Page 4, remove "statistically" in "there is no statistically significant
   increase in the performance".
8. It will be great if the computational CPU times for each optimizer can be
   reported, because it is often the computational burden that prohibits the
   application of the $L_0$ regularization.

**Strengths And Weaknesses:**

Strength:
1. The paper considers many real data sets in the comparisons, while existing
   works usually use single or a very small number of real data, or exclusively
   use synthetic data.
2. It uses multiple optimizers for both the $L_0$ and $L_1$ regularization,
   which is the first to consider the effect of the optimizers.

Weakness:
1. The presentation is a little chaotic, with many inconsistent notations. For
   example, it defines the acronym SNR for signal-to-noise ratio early on, but
   keeps switching between SNR and signal-to-noise ratio throughout the
   paper. This makes the paper hard to read, especially since it involves many
   other acronyms that do not have interpretations as clear as SNR, such as IHT,
   IHTM, SAGA, among others.
2. It also repeats the same content in different sections. For example, the
   first paragraph repeats the same content in the latter half of Section 2. It
   seems better to aggregate this paragraph into the latter half of
   Section 2. Actually, it would be better to move the latter half of Section 2
   into Section 3, because that part is talking about the optimizers used in the
   paper instead of "Related Studies" on comparing $L_0$ and $L_1$
   regularization.
3. I think Best Subset Selection and $L_0$ mean the same thing in the paper. If
   so, using $L_0$ is much better whenever possible. It is fine to mention Best
   Subset Selection a few times, but simultaneously using both throughout the
   paper just adds to the load for readers to follow the paper.

---

> ### Author Response · Authors · 2025-04-25
>
> 1. Thank you for pointing this out.
> We have made our use of SNR consistent in our revision.
> We have also added this to our revision to clarify the acronyms/names of optimizers:
>
> In this section, we present comparisons between variants of the Iterative Hard Thresholding (IHT) \citep{blumensath2009iterative}, with momentum (IHTM),
> and L0Learn \citep{dedieu2021learning,hazimeh2023l0learn}
> Best Subset Selection optimizers,
> the optimizer with mixed $L_0$ and $L_2$ penalty from L0Learn,
> and two LASSO optimizers: LIBLINEAR \citep{fan2008liblinear} and SAGA \citep{defazio2014saga}, which was based on Stochastic Average Gradient (SAG) method \citep{schmidt2017minimizing}.
>
> 2. Thank you for pointing this out.
> We have implemented your suggestion to move the descriptions of the optimizers into section 3 in our revision.
>
> 3. Yes, they do, thank you for pointing this out.
> Besides the title, introduction, and conclusion, we have changed 'best subset' and 'best subset selection' to '$L_0$-regularized' throughout the revised manuscript.
>
> 4. Thank you for pointing this out.
> We have added this statement to section 3 of our revision to clarify:
>
> The performance of the optimizers on the original datasets without added noise is displayed in each figure at the points for which $\sigma_X = 0$ and $p_y = 0$, which is when no feature noise or label noise is added.
>
> 5. Thank you for this concern.
> We have remade all the plots in our revision according to the requests of reviewer Vq3q, including shaded interquartile ranges.
> We have also removed the markers which were previously inconsistent.
>
> 6. Thank you - we have corrected the Fig 1 in the revision.
>
> 7. Thank you - it is consistent in the revision.
>
> 8. Thank you - it is defined in the revision.
>
> 9. You are correct - we have changed the statement in the revision. Thank you.
>
> 10. Yes, thank you for pointing this out. We have added Fig 5 and the accompanying discussion to our revision.

---

### Review · Reviewer_2YvM · 2025-04-01

**Summary Of Contributions:**

This paper investigates the classical variable selection problem by conducting a comparative study of various penalty terms and optimization algorithms. The penalties considered include $\ell_0$, $\ell_1$, and a combination of $\ell_0+\ell_2$, while the optimizers evaluated are IHT, IHTM, L0Learn, LIBLINEAR, and SAGA. These methods are benchmarked across several selected datasets.

**Audience:**

Yes

**Claims And Evidence:**

Yes

**Requested Changes:**

See "Strengths And Weaknesses"

**Strengths And Weaknesses:**

While this paper aims to provide a comprehensive study and analysis—as suggested by its title—I find that the scope and depth of the work fall short of this claim. The conclusions presented are either already well-established or lack sufficient depth. Therefore, it's difficult for researchers to extract interesting or useful insights from the study. Below are some specific concerns:

**(Algorithmic selection).** The selection of penalty terms and optimization algorithms is quite limited. For example, the widely-used elastic net penalty ($\ell_1 + \ell_2$) is omitted. Furthermore, the experiments include only three algorithms for $\ell_0$ penalties, one for $\ell_0 + \ell_2$, and two for $\ell_1$. Considering the extensive literature on variable selection, this set of methods does not support the claim of a “comprehensive comparison.” While a regular technical paper need not exhaustively benchmark against all existing methods, such breadth is expected when making a strong claim of comprehensiveness. In this context, the choices appear insufficient.

**(Presentation).** The paper lacks clarity in presenting both the algorithms and the datasets. Although IHT and IHTM are introduced with sufficient detail, other optimizers (e.g., L0Learn, LIBLINEAR, SAGA) are not clearly described. Similarly, the datasets are not properly characterized—important properties such as dataset size, feature range, and condition numbers are not reported. These factors are crucial for interpreting the performance results and understanding the practical implications of the study.

**(Well-known conclusions).** Regarding the analysis on $\ell_0$ vs $\ell_1$, most of the conclusions provided in this paper are already well-known. For example, (I) At low to moderate noise levels, $\ell_0$ regularization often outperforms $\ell_1$. This is because $\ell_0$ is an unbiased estimator, whereas $\ell_1$ introduces bias due to its shrinkage effect. While $\ell_1$ is more robust to noise owing to its convexity, in low-noise settings, both methods may correctly recover the support, with $\ell_0$ offering more accurate estimates due to its lack of bias. (II) $\ell_0$ regularization generally yields sparser solutions than $\ell_1$, as it directly penalizes the number of nonzero coefficients. In contrast, $\ell_1$ promotes sparsity indirectly, often resulting in models with more selected variables.

**(Lack of theoretical connections).** Since many of the conclusions in this paper are not novel, it is important to situate the findings within the context of existing literature—particularly papers that offer theoretical analysis. Drawing connections between the empirical results and established theoretical insights would help readers better understand not just what was observed, but why those observations occur.

**(Lack of in-depth analysis).** Most of the conclusions and observations in this paper are lacking deeper analysis or insight into why these patterns occur. For instance, the authors report that certain algorithms or penalty terms yield more accurate or sparser results, but do not explore the underlying reasons. The paper would be significantly strengthened by a deeper dive into specific datasets, providing more detailed observations and reasoning behind the results. For example, if an algorithm (say, Algorithm A) produces both sparse and accurate outcomes, it would be insightful to examine which features it selects or excludes—particularly if those excluded features correspond to noise. Such analysis could reveal why Algorithm A performs better and help explain the empirical trends observed. This level of detail is essential for drawing meaningful conclusions and offering practical guidance to researchers and practitioners.

---

> ### Author Response · Authors · 2025-04-25
>
> 1. Thank you for your comment. You are correct that we have not compared all possible forms of regularization penalty that can be found in the literature (for example, the elastic net).
> The scope of our paper is focused on comparing L0 and L1 methods and the effects of dataset noise in this comparison.
> Making this comparison systematic already requires significant computing resources.
> Each of the (6) optimizers we currently compare is run on 22 datasets, with 11 types of noise sampled randomly 20 times each, and each run through 50 trials of optuna for hyperparameter optimization. Altogether this means that every optimizer will be run more than 240,000 times, which occupies a significant fraction of our compute.
>
> To redress your concerns, we have added the following paragraph to our revision:
>
> Our focus is to compare $L_1$ to a method in $L_0$ selection as both are oriented towards some form of ``minimum subset'', and represent two widely studied and used approaches. While many interpolations of these exist, extensive numerical simulations require we limit ourselves from excessive additional penalties like the Elastic-Net (which intentionally will select correlated predictors) and $L_q$ penalties for $q \in (0, 1)$. We compare the L0Learn optimizer with both the pure best subset $L_0$ selector as well as a mixed selector which has both $L_0$ and $L_2$ penalties. We include the mixed $L_0$ and $L_2$ selector to make contact with the authors' existing study and recommendations.
>
> 2. Thank you for bringing these omissions to our attention. To redress your concerns, we have updated Table 1 to describe the datasets, and we have added the following texts describing the methods to our revision (partially omitted in this box due to character limits - please see pdf for full revision):
>
> The L0Learn optimizers use coordinate descent and local combinatorial search to approximately minimize its objective,
> %
> \begin{align}
> \min_{\boldsymbol{\theta}} & \quad f(\boldsymbol{\theta}) + \lambda_0 \|\boldsymbol{\theta}\|_0 + \lambda_2 \|\boldsymbol{\theta}\|_2,
> \end{align}
> %
> ...
>
> LIBLINEAR~\cite{fan2008liblinear} solves $L_1$-regularized logistic regression problems using coordinate descent, where the objective is
> %
> \[
> \min_{\boldsymbol{\theta}} \; \frac{1}{n} \sum_{i=1}^n \log\left(1 + \exp(-y_i \cdot x_i^\top \boldsymbol{\theta})\right) + \mathcal{R}(\boldsymbol{\theta}),
> \]
> %
> ...
>
> SAGA~\cite{defazio2014saga} is a variance-reduced stochastic gradient method designed for finite-sum problems. At each iteration, SAGA samples a data point \( i \) uniformly at random and computes the gradient of the loss \( f_i(\boldsymbol{\theta}_t) \) on that example. It corrects this gradient using stored gradients from previous iterations to reduce variance, resulting in the update:
> %
> \[
> \boldsymbol{\theta}_{t+1} = \operatorname{prox}_{\eta}^{\mathcal{R}} \left( \boldsymbol{\theta}_t - \eta \left( \nabla f_i(\boldsymbol{\theta}_t) - \nabla f_i(\boldsymbol{\phi}_i) + \bar{\boldsymbol{\alpha}} \right) \right),
> \]
> ...
>
> 3. You are correct that for data generated by a linear model with a fixed number of nonzero weights, the best subset selector is an unbiased estimator. As you said, this was long known. The focus of this manuscript is not the performance of using sparse logistic regression on simulated data with known data-generating functions, but on real-world noisy data with unknown data-generating functions that are likely not linear. In these cases, both the $L_0$ and $L_1$ selectors are likely biased.
> Additionally, large performance differences between different $L_0$ optimizers on binary classification tasks have not been previously studied.
>
> 4. We have endeavored to connect to the existing theoretical works that performed studies on synthetic datasets in our related works section. You are correct that uncovering theoretical tools that are predictive of out-of-sample generalization of sparse logistic regression models on real-world data with unknown and potentially non-linear data-generating functions would be valuable but a significant undertaking with a greatly enlarged scope.
>
> 5. We agree that would be useful but also a significant undertaking. Please see our response to your previous point.

---

### Author Response · Authors · 2025-04-07
**Additional Clarifying Experiments**

Hello,
We thank the reviewers for their feedback; incorporating the feedback will greatly strengthen and clarify the paper.
We are currently running additional experiments to properly and thoroughly address the reviewer's comments and questions.
We would ask for an additional 1-2 weeks for our response.

Thank you

---

> ### Comment · Reviewer_Vq3q · 2025-04-22
>
> Can the authors please respond to the reviews within the next week? I would like my final recommendation to take into account the authors comments and any new experiments that they have prepared.

---

> ### Author Response · Authors · 2025-04-22
>
> Yes, we will respond within the week.
> Thank you.

---

### Decision · Action_Editor_vEom · 2025-05-08

**Recommendation:** Reject

**Comment:**

The paper received detailed feedback from three reviewers who provided a consistent set of critical points:

Reviewer 2YvM emphasized concerns about novelty, asserting that many conclusions (e.g., L0 models producing sparser but potentially less robust results under higher noise scenarios) are already well-documented in the literature. They also noted the scope of methods considered insufficiently comprehensive, particularly highlighting the absence of elastic-net regularization and other widely-used methods.

Reviewer FNTA criticized the clarity and presentation quality, including inconsistent acronym usage and unclear explanations of algorithms and datasets. Although the authors claimed improvements, the reviewer found these concerns inadequately addressed.

Reviewer Vq3q raised substantial methodological concerns. They criticized the black-box hyper-parameter optimization approach, inconsistent and optimized termination criteria for each optimizer, and the lack of exact solutions as benchmarks for L0 methods. The reviewer argued these issues rendered the reported findings unreliable and uninterpretable. Despite the authors' clarification and attempts to position the paper as practitioner-oriented, Reviewer Vq3q argued that early stopping with extensively tuned termination conditions does not yield results representative of standard L0 or L1 regularized problems.

The authors made attempts to respond to these critiques by clarifying the scope, modifying the title to explicitly include "with Early Stopping," and adjusting their claims to emphasize the empirical, approximate nature of their comparisons. Despite these changes, the reviewers maintained their positions, primarily due to fundamental concerns around experimental methodology and the interpretability of the findings.

In summary, while the submission addresses an important topic and presents extensive experimentation, methodological and presentation concerns raised by the reviewers remain unresolved. As such, I support the reviewers' unanimous recommendation to reject the paper in its current form.

**Audience:**

While the general topic of sparse learning in classification is relevant to the TMLR audience, reviewers consistently felt that the contribution, in its current form, lacked depth and clarity. Although some readers interested in practical aspects of sparse learning might find value in the extensive empirical results, the issues raised regarding methodology and presentation significantly diminish the broader appeal and usefulness of the paper's conclusions to the TMLR audience.

**Claims And Evidence:**

The submission presents an empirical study comparing L0 and L1 sparse learning approximations with early stopping across various binary classification datasets. However, there are significant concerns regarding whether the claims made are supported convincingly by the presented evidence. Specifically, reviewers highlighted that the extensive hyper-parameter tuning and varying optimizer termination conditions significantly complicate interpreting the results. Due to these methodological issues, the reported performance differences between optimization methods may reflect optimizer-specific behaviors and tuning strategies rather than intrinsic properties of the L0 or L1 regularization techniques themselves. Furthermore, the absence of an exact solver for the L0 penalized problems prevents a definitive evaluation of the approximate solutions provided by IHT and L0Learn.

Thus, the claims regarding the superiority or competitiveness of L0 penalization methods with early stopping relative to L1 methods are not convincingly substantiated by the current experimental setup.